# Invited perspectives: Landslide populations - can they be predicted?

Fausto Guzzetti [1,2]

(1) Dipartimento della Protezione Civile, Presidenza del Consiglio dei Ministri, via Vitorchiano 2, 00189 Roma

(2) Istituto di Ricerca per la Protezione Idrogeologica, Consiglio Nazionale delle Ricerche, via della Madonna Alta 126, 06129 Perugia

Landslides are different from other natural hazards. Unlike volcanoes, they do not threaten human civilization (Papale and Marzocchi, 2019). Unlike tsunamis, they do not affect simultaneously several thousands of kilometres of coastline – although a submarine landslide in Norway has caused a tsunami to hit Scotland (Dawson et al., 1988). Unlike floods and earthquakes, they do not cause hundreds of thousands of casualties in a single event – although a landslide has killed thousands in Peru (Evans et al., 2009) and debris flows tens of thousands in Colombia (Wieczorek et al., 2001). But the human toll of landslides is high (Froude and Petley, 2018), and their economic and societal consequences are largely undetermined. Compared to other hazards, landslides are subtle, often go unnoticed, and their consequences are underestimated.

As with other hazards, the design and implementation of effective risk reduction strategies depend on the ability to predict (forecast, project, anticipate) landslides. I have argued that "*our ability to predict landslides and their consequences measures our ability to understand the underlying […] processes that control or condition landslides, as well as their spatial and temporal occurrence*" (Guzzetti, 2021). This assumes that landslide prediction is possible; something that has not been demonstrated (or disproved), theoretically. Yet, there is nothing in the literature that prevents landslide prediction; provided that one clarifies the meaning of "prediction" (Guzzetti, 2021), that the prediction is scientifically based (Guzzetti, 2015), and that we understand the limits of the prediction (Wolpert, 2001). Efforts are needed to determine the limits of landslide predictions, for all landslide types (Hungr et al., 2014) and at all geographic and temporal scales (**Figure 1**).

Here, I outline what I consider to be the main problems that need to be addressed in order to
advance our ability to predict landslide hazards and risk. The field is vast, and I limit my perspective
to populations of landslides – that is, the hazards and risk posed by many landslides caused by one
triggering event, or by multiple events in a short period. In this context, predicting landslide hazard
means anticipating *where*, *when*, *how frequently*, *how many*, and *how large* populations of
landslides are expected (Guzzetti et al., 2005; Lombardo et al., 2020; Guzzetti, 2021). Predicting
landslide risk is about anticipating the consequences of landslide populations to different
vulnerable elements (Alexander, 2005; Glade et al., 2005; Galli and Guzzetti, 2007; Salvati et al.,
36 2018).

Landslides tend to occur where they have previously occurred (Temme et al., 2020). Therefore,
one way to assess *where* they are expected is to map past and new landslides. The technology is
mature for regional and even global landslide detection and mapping services based on the
automatic or semi-automatic processing of aerial and satellite imagery; optical, SAR and LiDAR
data (Guzzetti et al., 2012; Mondini et al., 2021). An alternative – and complementary – way is
through susceptibility modelling; an approach for which there is no shortage of data-driven
methods, but rather of suitable environmental and landslide data (Reichenbach et al., 2018). The
increasing availability of satellite imagery, some of which repeated over time and free of charge
(Aschbacher, 2017), opens unprecedented opportunities to prepare event and multi-temporal
inventory maps covering very large areas, which are essential to build space-time prediction models
(Lombardo et al., 2020), to investigate the legacy of old landslides on new ones (Samia et al., 2017;
Temme et al., 2020), to obtain accurate thematic data for susceptibility modelling (Reichenbach et
al., 2018), and to validate geographical landslide early warning systems (Piciullo et al., 2018;
Guzzetti et al., 2020). However, the literature reveals a systematic lack of standards for
constructing, validating, and ranking the quality of landslide maps and prediction models (Guzzetti
et al., 2012; Mondini et al., 2021; Reichenbach et al., 2018). This reduces the credibility of the
maps and models. A gap that urgently needs to be bridged (Guzzetti, 2021).
Predicting *when* or *how frequently* landslides will occur can be done for short and for long periods.
For short periods – from hours to weeks – the prediction is obtained through process-based models,
rainfall thresholds, or their combination. Process-based models rely upon the understanding of the
physical laws controlling the slope instability conditions of a landscape forced by a transient trigger
e.g., a rainfall, snow melt, seismic, or volcanic event (Bogaard and Greco, 2016, 2018). The major
limitation of physically-based models is the scarcity of relevant data, which are hard to obtain for
very large areas. New approaches to obtain relevant, spatially-distributed data are needed, as well
as novel models able to extrapolate what is learned in sample areas to vast territories (Bellugi et
al., 2011; Alvioli and Baum, 2016; Alvioli et al., 2018; Mirus et al., 2020).
Thresholds are empirical or statistical models that link physical quantities (e.g., cumulative rainfall,
rainfall duration) to the occurrence – or lack of occurrence – of known landslides. Reviews of the
literature (Guzzetti et al., 2008; Segoni et al.; 2018) have highlighted conceptual problems with the
definition and use of rainfall thresholds for operational landslide forecasting and early warning
(Piciullo et al., 2018; Guzzetti et al., 2020), including the lack of standards for defining the
thresholds and their associated uncertainty (Melillo et al., 2018), and for the validation of the
threshold models (Piciullo et al., 2017, 2018; Guzzetti et al., 2020). The community needs shared
criteria and algorithms coded into open-source software for the objective definition of rainfall
events, of the rainfall conditions that can result in landslides, of rainfall thresholds (Melillo et al.,
2015, 2018), and for the validation of the threshold models (Piciullo et al., 2017). This will not
only provide reliable and comparable thresholds, allowing for regional and global studies (Guzzetti
et al., 2008; Segoni et al.; 2018), but also increase the credibility of early warning systems based
on rainfall threshold models (Guzzetti et al., 2020).
The projection of landslide frequency for long periods – decades to millennia – is much more
difficult and uncertain, as it depends on climatic and environmental characteristics that are poorly
know and difficult to measure and model (Crozier, 2010; Gariano and Guzzetti, 2016), as well as
on the inherent incompleteness of the historical landslide records (Rossi et al., 2010). The literature
on the analysis of historical landslide records remains scarce, but the number of studies projecting
the future occurrence of landslides is increasing (Gariano et al., 2017; Peres and Cancelliere, 2018;
Schlögl and Matulla, 2018; Patton et al., 2019; Schlögel et al., 2020; Gariano and Guzzetti, 2021).
In this field, studies will be relevant if they compare analyses and validation methods in different
areas. This requires the exchange of data and information.
Predicting *how many* and *how large* landslides are expected means anticipating the size (e.g., area,
volume, length, width, depth) and number of landslides in an area – with size and number correlated
in a population of landslides. This information is obtained by constructing and modelling
probability distributions of landslide sizes obtained typically from landslide event inventory maps
(Stark and Hovius, 2011; Malamud et al., 2004). The literature on the topic is limited, and with
differences in the way the distributions are modelled. This hampers comparisons from different
areas. Although models have been proposed to explain the probability size distributions (Katz and
Aharanov, 2006; Stark and Guzzetti, 2009; Klar et al., 2011; Bellugi et al., 2021), further efforts
are needed to explain the observed distributions of landslide sizes, and to evaluate their variability
and uncertainty.
By combining probabilistic information on *where*, *when* or *how frequently*, and *how many* or *how*
*large* landslides are, one can evaluate landslide hazards for different landslide types. However, the
existing models are crude, they work under assumptions that are difficult to prove (Guzzetti et al.,
2005), and the possibility to export them in different areas is limited, or untested. Novel efforts are
needed to prepare reliable landslide hazard models (Lombardo et al., 2020; Guzzetti, 2021).
Assessing landslide hazard is important but, for social applications what is needed is the estimation
of the landslide consequences, which means assessing the vulnerability to landslides of various
elements at risk (Alexander 1999; Galli and Guzzetti 2007), and evaluating landslide risk (Cruden
and Fell, 1997; Glade et al., 2005; Porter and Morgenstern, 2013), including risk to the population
(Petley, 2012; Froude and Petley, 2018; Salvati et al., 2018; Rossi et al., 2019). Here, the main
limitation is the difficulty to obtain data on landslide vulnerability, and reliable records of landslide
events and their consequences (Petley, 2012; Froude and Petley, 2018; Salvati et al. 2018). Where
the information is available, comprehensive landslide risk models can be constructed, and validated
(Rossi et al., 2019). It is important that efforts are made to collect reliable records of landslides and
their consequences, and that the records are shared to test different risk models.
Of the various factors governing landslide hazard the most uncertain and the one requiring more
urgent efforts is the time prediction (*when*, *how frequently*), followed by the prediction of the size
and number of expected failures. For both, multi-temporal inventories and landslide catalogues are
essential to build innovative predictive models. To construct the records, systematic efforts are
needed for landslide detection and mapping (Mondini et al., 2021). For susceptibility (*where*), the
challenge is to prepare reliable regional, continental, or global assessments (Stanley and
Kirschbaum, 2017; Broeckx et al., 2018; Wilde et al., 2018; Mirus et al., 2020). Critical are also
novel modelling frameworks combining the hazard factors (Lombardo et al., 2020). But the goal
is to reduce risk (Glade et al., 2005). For that, vulnerability studies, improved early warning
capabilities, quantification of the benefits of prevention, and better risk communication strategies
are crucial (Guzzetti, 2018). Much work is needed on these largely unexplored subjects.
Ultimately, I note that in medicine – a field of science conceptually close to the field of landslide
hazard assessment and risk mitigation (Guzzetti, 2015) – the paradigm of "convergence research"
is emerging (Sharp and Hockfield, 2017), where *"convergence comes as a result of the sharing of*
*methods and ideas … It is the integration of insights and approaches from historically distinct*
*scientific and technological disciplines"* (Sharp et al., 2016). The community of landslide scientists
should embrace the paradigm of "converge research", exploiting the vast amount of data,
measurements, and observations that are available and will be collected, expanding the making and
use of predictions, assessing the economic and social costs of landslides, designing sustainable
mitigation and adaptation strategies, and addressing the ethical issues posed by natural hazards,
including landslides (Bohle, 2019). I am persuaded that this will contribute to advancing
knowledge and building a safer society (Guzzetti, 2018).

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

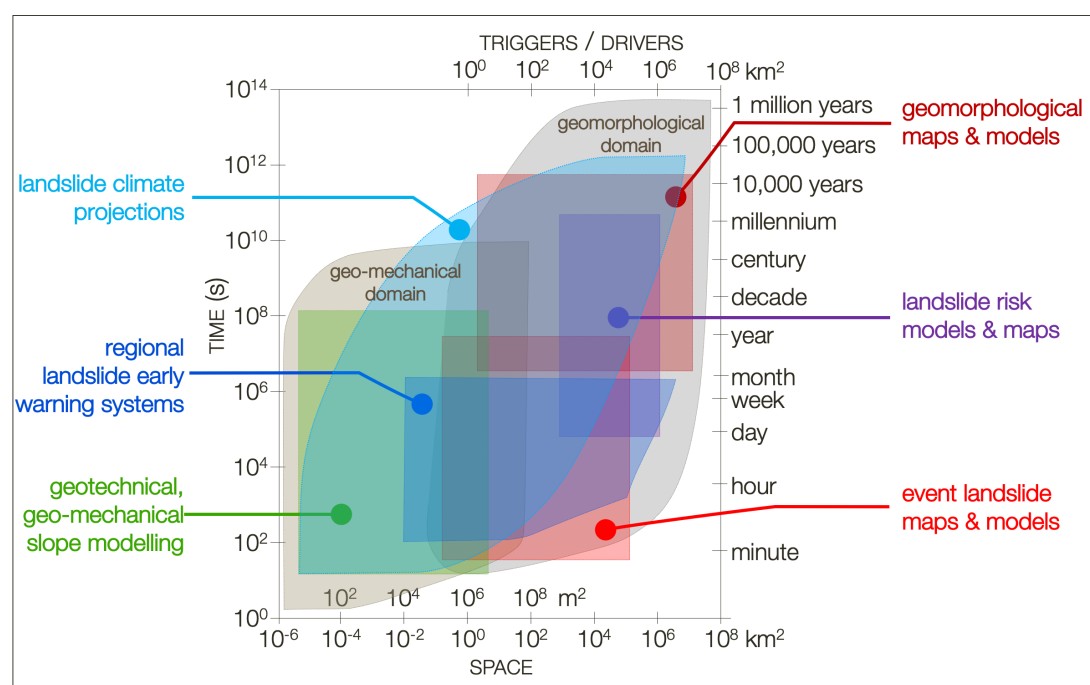

Figure 1. Space (lower x-axis) – time (y-axes) chart showing main geomorphological and geo-mechanical landslide domains, and typical length-scale of main meteorological and geophysical triggers and drivers of populations of landslides. Coloured polygons show approximate sub-domains for typical landslide hazards and risk mapping and modelling efforts. Modified after Guzzetti (2021).