# Peer review of "Invited perspectives: Landslide populations - can they be predicted?"

_Natural Hazards and Earth System Sciences, 2021_

## Author Response (AR2)

**Comments of the editor**

*Dear Fausto Guzzetti,*

*Thank you for the submission of your very interesting manuscript "Invited perspectives: Landslide populations – can they be predicted?".*

*As you know, two reviewers have provided good and interesting suggestions on how to improve your manuscript, which you have replied to. Both reviewers recommended minor revisions, and therefore I would like to invite you to submit a revised version of your manuscript.*

*In view of the suggestions from the reviewers and your response, I would like to stress the following: My main and first priority is to publish an as valuable and as interesting "invited perspectives" as possible. My minor concern is the length or number of words necessary to present the perspective. From my point of few, the manuscript should be concise and not unnecessary lengthy (from which it is far away), but in case one or two pages more are necessary to provide some more information and particularly ideas for perspective and the way forward, this is not a problem. I also think, that it is good to have a figure included, so I suggest not to delete it.*

*I would like to particularly support the suggestion of referee #1: "Since this is an invited perspective, I suggest to elaborate more about the next steps and the way forward, in respect to what is needed from science, what scientific suggestions do you have, what approaches should be followed, which studies should be undertaken.*

*I look forward to seeing the next version of your manuscript which I will not send out for further review, but rather, will make the decision myself, assuming no major items come up in the revised manuscript for which I need outside reviewers to aid me in my decision.*

*Best regards*

*Heidi Kreibich*

*NHESS executive editor*

**Response to the comments of the editor**

Dear Heidi,

Thank you for your comments and valuable recommendations on the first submission of my article.

Based on your recommendations and those of the two reviewers, I have prepared a new version of the article, which I submit for possible publication in the special issue of NHESS you are preparing.

The new version of the article is longer than the previous version (1746 words, excluding title, author, affiliations, references, and figure caption). I have accepted your recommendation to add relevant information. Nevertheless, I have tried to keep the overall article short, in the spirit of the special issue.

Following your recommendation and a similar recommendation of the first reviewer, for each of the main sections of the text I have added language to explain more clearly – albeit always shortly – what are the main efforts that are needed, in my opinion, to improve the existing landslide prediction capabilities. I have also clarified the problems and inherent limitations of the various predictions (where, when, how many / how large, etc.). Throughout the article, most of the new text was added for this scope.

Following the recommendation of one of the referees, I have added an entire new paragraph discussing what are the efforts that I consider more (and less) urgent for improved landslide hazard and risk prediction. The new text, towards the end of the article, reads "Of the various factors governing landslide hazard the most uncertain and the one requiring more urgent efforts is the time prediction (when, how frequently), followed by the prediction of the size and number of expected failures. For both, multi-temporal inventories and landslide catalogues are essential to build innovative predictive models. To construct the records, systematic efforts are needed for landslide detection and mapping (Mondini et al., 2021). For susceptibility (where), the challenge is to prepare reliable regional, continental, or global assessments (Stanley and Kirschbaum, 2017; Broeckx et al., 2018; Wilde et al., 2018; Mirus et al., 2020). Critical are also novel modelling frameworks combining the hazard factors (Lombardo et al., 2020). But the goal is to reduce risk (Glade et al., 2005). For that, vulnerability studies, improved early warning capabilities, quantification of the benefits of prevention, and better risk communication strategies are crucial (Guzzetti, 2018). Much work is needed on these largely unexplored subjects."

The section on the temporal prediction of landslides (*when* or *how frequently*) is now longer, and more detailed. I have added language to discuss the use of physically-based models, their limitations, and the enhancements needed. The new text reads "The major limitation of physically-based models is the scarcity of relevant data, which are hard to obtain for very large areas. New approaches to obtain relevant, spatially-distributed data are needed, as well as novel models able to extrapolate what is learned in sample areas to vast territories (Bellugi et al., 2011; Alvioli and Baum, 2016; Alvioli et al., 2018; Mirus et al., 2020)".

In addition, the section now contains a description of the enhancements suggested to improve the definition of rainfall threshold models. The new text reads "The community

needs shared criteria and algorithms coded into open-source software for the objective definition of rainfall events, of the rainfall conditions that can result in landslides, of rainfall thresholds (Melillo et al., 2015, 2018), and for the validation of the threshold models (Piciullo et al., 2017). This will not only provide reliable and comparable thresholds, allowing for regional and global studies (Guzzetti et al., 2008; Segoni et al.; 2018), but also increase the credibility of early warning systems based on rainfall threshold models (Guzzetti et al., 2020)".

Following the recommendation of one of the referees, I have modified the last paragraph, to explain how the adoption of the "converge research" paradigm can help improving landslide hazards and risk modelling and prediction. The new text reads "The community of landslide scientists should embrace the paradigm of "converge research", exploiting the vast amount of data, measurements, and observations that are available and will be collected, expanding the making and use of predictions, assessing the economic and social costs of landslides, designing sustainable mitigation and adaptation strategies, and addressing the ethical issues posed by natural hazards, including landslides (Bohle, 2019). I am persuaded that this will contribute to advancing knowledge and building a safer society (Guzzetti, 2018)".

Following your recommendation, I have left the Figure (Figure 1) and its captions. The Figure is now cited earlier on in the text.

Where I explain how the landslide spatial prediction ("where") is made, and how it can be improved, I have reversed the logic of the text, discussing first landslide detection and mapping, and then landslide susceptibility modelling. To abide to the request of one of the reviewers, I have added language to explain the types of remote sensing images used for landslide detection and mapping (e.g., optical, SAR, LiDAr).

Since there is no limit to the number of references, I have added quit a few of them, 27 in total. This has increased the length of the main text, due to the citations. Regarding the list of references, I have checked the style and made a number of changes and adjustments to be compliant with the journal formatting rules.

Ultimately, to reduce the sections of text identified in the "similarity report", I have checked the text, and I have changed / rephrased I, wherever this was possible.

Most of the changes and additions made to the text are outlined in red in the file: Text-r2v02-20210406-changes.docx.

Below, I include my comments and responses to the two anonymous reviewers.

I hope that in the present, revised form the manuscript is acceptable for publication in the special issue of NHESS.

I look forward to hearing your editorial decision.

Kind regards,

Fausto Guzzetti

**Response to the comments of the first reviewer**

*The manuscript "Invited perspectives: Landslide populations - can they be predicted?" provides a very interesting concise review about what approaches and data is available for predicting hazard and risk of populations of landslides. The manuscript is very well structured into the individual questions: where? when or how frequently? how many and how large? Consequences?*

*From my point of view this is valuable, however, with adding some more information and particularly ideas for perspective and the way forward, this manuscript can become significantly more interesting for the scientific community.*

I am pleased that the reviewer has found the manuscript valuable, and I thank the reviewer for the positive comments and the useful recommendations.

*Thus, I suggest the following:*

*For each of the tackled questions knowledge gaps are presented. It would be great to put this somehow in order in respect to the urgency they should be closed. Please add some more information on which problems are important to solve (and maybe which are not so important but scientifically interesting). Please add one paragraph on what is the most pressing knowledge gap or challenge in respect to landslide predictions. Elaborate why it is important to solve this problem, if possible link this to practical problems of civil protection.*

As I have written in a previous post in response to the comment of this reviewer, the Editors have requested a (very!) short article, which will be part of a special issue celebrating the 20th anniversary of the journal. The limits in length, make it very difficult to respond fully to the request of the reviewer. As requested, I have added a new paragraph in the text. The new language reads "Of the various factors governing landslide hazard the most uncertain and the one requiring more urgent efforts is the time prediction (when, how frequently), followed by the prediction of the size and number of expected failures. For both, multi-temporal inventories and landslide catalogues are essential to build innovative predictive models. To construct the records, systematic efforts are needed for landslide detection and mapping (Mondini et al., 2021). For susceptibility (where), the challenge is to have reliable regional, continental, or global assessments (Stanley and Kirschbaum, 2017; Broeckx et al., 2018; Wilde et al., 2018; Mirus et al., 2020). Critical are also novel modelling frameworks combining the hazard factors (Lombardo et al., 2020). But the goal is to reduce risk (Glade et al., 2005). For that, vulnerability studies, improved early warning capabilities, quantification of the benefits of prevention, and better risk communication strategies are crucial (Guzzetti, 2018)."

*Since this is an invited perspective, I suggest to elaborate more about the next steps and the way forward, 1) in respect to what is needed from science, what scientific suggestions do you have, what approaches should be followed, which studies should*

*be undertaken. At the end "convergence research" is mentioned, please elaborate more on how this could look like in respect to landslide prediction, what would be necessary to undertake convergence research of landslide prediction, etc. and 2) what are practical solutions, what can be done by practitioners in respect to landslide prediction, what are practical steps forward.*

Considering the mentioned restrictions imposed by the editors on the length of the article, I have done my best to address the issues raised by the reviewer adding language to the last paragraph of the article, which nor reads: "Ultimately, I note that in medicine – a field of science conceptually close to the field of landslide hazard assessment and risk mitigation (Guzzetti, 2015) – the paradigm of "convergence research" is emerging (Sharp and Hockfield, 2017), where "convergence comes as a result of the sharing of methods and ideas … It is the integration of insights and approaches from historically distinct scientific and technological disciplines" (Sharp et al., 2016). The community of landslide scientists should embrace the paradigm of "converge research", exploiting the vast amount of data, measurements, and observations that are available and will be collected, expanding the making and use of predictions, assessing the economic and social costs of landslides, designing sustainable mitigation and adaptation strategies, and addressing the ethical issues posed by natural hazards (Bohle, 2019). This will contribute to advancing knowledge, and building a safer society (Guzzetti, 2018)."

*It would be interesting to know what this all means for practice, e.g. the Italian Civil Protection. How severe is the landslide risk in Italy? What are the main challenges and problems Italian Civil Protection is facing in respect to landslides? How do they approach the landslide risk and what are they planning to do about it in the future? Please add at least one paragraph about these practical aspects.*

I have considered this comment thoroughly and, in the end, I have decided not to discuss the issues posed by the reviewers; for two main reasons. First, the paper is meant to be general, and not focused on Italy and on what the Italian Civil Protection does to address landslide risk in Italy. Second, an – even very synthetic – discussion of the topic will require quite a bit of text, which is not available given the format of the paper.

**Response to the comments of the second reviewer**

*I read with interest this invited perspective that is concise, clear, and well-focused in its purpose. The article is aligned with the goals of the 20th anniversary of the journal Natural Hazards and Earth System Sciences (NHESS) Special Issue. The discussed topic is relevant.*

I am pleased that the reviewer has found the manuscript of interest and the topic relevant, and I thank the reviewer for the positive comments and the useful recommendations.

*However, there are few minor points that could be improved, and therefore help the readers to better catch the message of the entire work. I summarize these in the following points:*

I respond to the comments of the reviewer below.

*"the point of view of the organization", in this case, "Dipartimento della Protezione Civile" of Italy, should be further described, with just 2-3 more sentences (but we cannot enlarge too much the work giving the SI purpose of two-pages limit) on the operational point of view of landslide prediction & mitigation framework. The authors are curious to see and learn from the Italian case study.*

I have considered this comment thoroughly and, I have decided not to discuss the point of view of the Italian "Dipartimento della Protezione Civile" of Italy" for two reasons. First, the paper is meant to be general, and not focused on Italy and on what the Italian Civil Protection does – or does not – to address landslide risk in Italy. Second, I am not really entitled to provide the official perspective of the "Dipartimento della Protezione Civile".

*Line 42: when discussing "remote sensing imagery" I suggest mentioning the name (within brackets) of the remote sensing technologies (which are commonly used in preparing landslide maps).*

I have edited the text as requested. The new text reads "The increasing availability of remote-sensing imagery (optical, SAR, LiDAR), some of which is repeated over time and free of charge (Aschbacher, 2017), opens …".

*Line 63-64: the authors indicated that the number of studies projecting the future occurrence of landslides is increasing; here few key citations are necessary (we can exceed with citations since they are not included in the two-pages limit).*

I have added the requested references. The new text reads "The literature on the analysis of historical landslide records remains scarce (Rossi et al., 2010), but the number of studies projecting the future occurrence of landslides is increasing (Gariano et al. 2017; Peres and Cancelliere, 2018; Schlögl and Matulla, 2018; Patton et al. 2019; Schlögel et al. 2020; Gariano and Guzzetti, 2021)".

*Line 77, and in general in the entire work: the landslide types are not discussed or classified; this could be an interesting point for NHESS readers, but I understand that giving the limit of 2 pages and the focus of the work, maybe it is difficult to address. However, I would suggest thinking if there is a possibility to speculate a little on this, simply mentioning some landslide type when citing the literature.*

As the referee has indicated, it would have been very difficult to address the many issues related to the prediction of the hazards posed by different landslide types in a very short article. However (a) I maintain that the content of the manuscript is general and applies to most landslide types; and (b) I now specify in the text that the manuscript deals with all landslide types.

**Additional remarks**

I have added 18 new references.

I have corrected the way the list of references was written.

To compensate for a slightly longer text, due to the addition of an entire new paragraph requested by one reviewer, of additional lines of text, and of new citations, I have deleted Figure 1, which was not necessary.

I have checked the language of the manuscript.